# Multiscale Structural Insight into Dairy Products and Plant-Based Alternatives by Scattering and Imaging Techniques

**DOI:** 10.3390/foods12102021

**Published:** 2023-05-17

**Authors:** Theresia Heiden-Hecht, Baohu Wu, Marie-Sousai Appavou, Stephan Förster, Henrich Frielinghaus, Olaf Holderer

**Affiliations:** 1Jülich Centre for Neutron Science (JCNS) at Heinz Maier-Leibnitz Zentrum (MLZ), Forschungszentrum Jülich GmbH, Lichtenbergstr. 1, 85747 Garching, Germany; ba.wu@fz-juelich.de (B.W.);; 2Jülich Centre for Neutron Science (JCNS-1), Forschungszentrum Jülich GmbH, 52425 Jülich, Germany

**Keywords:** dairy products, cheese, plant-based emulsions, scattering, electron microscopy, SAXS

## Abstract

Dairy products and plant-based alternatives have a large range of structural features from atomic to macroscopic length scales. Scattering techniques with neutrons and X-rays provide a unique view into this fascinating world of interfaces and networks provided by, e.g., proteins and lipids. Combining these scattering techniques with a microscopic view into the emulsion and gel systems with environmental scanning electron microscopy (ESEM) assists in a thorough understanding of such systems. Different dairy products, such as milk, or plant-based alternatives, such as milk-imitating drinks, and their derived or even fermented products, including cheese and yogurt, are characterized in terms of their structure on nanometer- to micrometer-length scales. For dairy products, the identified structural features are milk fat globules, casein micelles, CCP nanoclusters, and milk fat crystals. With increasing dry matter content in dairy products, milk fat crystals are identified, whereas casein micelles are non-detectable due to the protein gel network in all types of cheese. For the more inhomogeneous plant-based alternatives, fat crystals, starch structures, and potentially protein structures are identified. These results may function as a base for improving the understanding of dairy products and plant-based alternatives, and may lead to enhanced plant-based alternatives in terms of structure and, thus, sensory aspects such as mouthfeel and texture.

## 1. Introduction

Food systems such as dairy products and plant-based alternatives are evaluated and defined, among other properties, by their texture, mouthfeel, taste, and flavor. In particular, the texture, structure, and mouthfeel are related to the structural characteristics of these food products. The structural characteristics of food are usually determined via (a) flow properties captured by rheological techniques, including viscosimeters and texture analyzers [1,2]; (b) friction properties analyzed by tribological techniques [1,2]; and (c) sensory properties [1,3]. However, the listed properties are linked to the structural characteristics, but do not characterize the multiscale structural properties. Details of structural properties may be analyzed via scattering techniques, such as small-angle X-ray or neutron scattering, or imaging techniques for a defined and finite area.

Previous publications identified the potential for imaging and scattering methods to characterize structural insights. Imaging and scattering methods usually identify structures from the micrometer to the nanometer range in situ, either without breakage or with minimal breakage of the structure [4]. For dairy components, numerous publications have characterized single components, such as casein micelles [4,5,6,7,8,9,10] and fat globules [4,11], and their crystallization behavior [4,12,13] mostly via small-angle X-ray, neutron scattering, or confocal laser scanning microscopy. For dairy products, many studies have focused on analyzing the structure of milk via scattering techniques [14,15,16] and the microstructure of coagulated milk via small-angle X-ray and neutron scattering [17]. The structure of yogurt and cheese [18,19] was mostly analyzed via imaging techniques [20,21]. For plant-based alternatives, fewer studies have been published that use imaging techniques. The structure of yogurt-like systems made from oat or pea protein was analyzed by imaging techniques [22,23,24], as was the structure of plant-based cheese [25,26,27]. However, the utilized imaging techniques, such as confocal laser scanning or scanning electron microscopy, using either staining or drying steps affected the analyzed structures, whereby an artefact-free image may be gained via ESEM [21,25,26].

To the best of our knowledge, a structure-based comparison of dairy products and plant-based alternatives was not performed for commercially available products. Therefore, we aim to compare the structure of common dairy products and plant-based alternatives with increasing dry matter content via SAXS for structures from 1 nm to 1 μm, and via ESEM for structures from 0.5 to 5 μm. By doing so, we aim to deliver the first database of common dairy products and plant-based alternatives investigated via SAXS and ESEM. This data base will lead to an improved understanding of the milk systems in a length scale range from 1 nm to 5 μm, and may function as a starting point to improve plant-based alternatives with the help of emerging techniques, including ESEM and SAXS, since the structure affects sensory aspects such as mouthfeel and the texture of food products.

Therefore, we investigated the multiscale structural insights of dairy products and plant-based alternatives via small-angle X-ray scattering (SAXS) and environmental scanning electron microscopy (ESEM) without affecting the structure. For dairy products and plant-based alternatives, at least one liquid emulsion with <12% d.m., one fermented emulsion with low dry matter content (like yogurt with <30% d.m.), and one emulsion with high dry matter content (like cheese with >40% d.m.) were chosen and analyzed via SAXS and ESEM.

## 2. Materials and Methods

Samples investigated in this paper are food emulsions and products derived from food emulsions. All samples are commercially available and were purchased from food distributors. The samples are categorized into liquid emulsions, fermented emulsions with low dry matter content, and fermented emulsions with high dry matter content. For each of these categories, at least one dairy product and one plant-based alternative were investigated, as summarized in Table 1. Dairy products originate from cow’s milk. The dairy products are further described with the EAN code, which is linked to the milk’s origin. Additional information about the products may be gained with these codes. The products with increasing dry matter content are as follows: UHT milk with 3.5% fat (AT 40138 EG, EAN: 4311501489840) as a liquid emulsion; yogurt with 3.5% fat (DE BY718 EG, EAN: 4008452011007) and cream cheese with 17% fat (DK M198 EC, EAN: 5711953103780) as fermented emulsions with low dry matter content; and Camembert (FR 88.470.001 CE, FR 49.107.001 CE EAN: 3161710000678), Gouda (DE BY40390 EG, EAN: 4311501356838), and a hard cheese (Parmigiano Reggiano, IT Y8020 CE, EAN: 4311501797310) as fermented emulsions with high dry matter content. Detailed information about the fat, carbohydrate, protein, and salt composition of all introduced samples may be found in Table 1. This information is summarized in the product declaration.

The plant-based liquid emulsions with increasing dry matter content are as follows: a plant-based drink with 3.5% fat, based on oat, pea protein, and sunflower oil (drink1, EAN: 5411188134985); and a plant-based drink with 3.0% fat, based on oat and rapeseed oil (drink2, EAN: 7394376616501). The plant-based fermented emulsions with low dry matter content are as follows: a yogurt-like fermented product with 1.9% fat, based on soy and oat, fermented with L. acidophilus, L. bulgaricus, and Str. thermophilus (fermented1-Y, EAN: 5411188130741); a cream cheese-like product based on coconut oil, starch, and gluco-delta-lacton (fermented2-CC, EAN: 5202390021121); and a cream cheese-like product based on rapeseed oil, coconut oil, oat, potato starch, and protein (fermented3-CC, EAN: 7394376617034). The plant-based fermented emulsions with high dry matter content are as follows: a cheese-like product based on coconut oil, starch, and gluco-delta-lacton (fermented1-C, EAN: 5202390020407); and a cheese-like product based on walnuts, coconut oil, starch, and potato protein (fermented2-C, EAN: 426444961015). Detailed information about the fat, carbohydrate, protein, and salt composition of all introduced samples may be found in Table 1.

All the listed samples (Table 1) were analyzed via SAXS to characterize the structural properties over a large range of length scales in reciprocal space. ESEM was used for real-space images of the chosen samples.

### 2.1. ESEM

Topography and structural alterations of dairy products and plant-based alternatives were observed with a FEI Quattro S (FEI Company, Eindhoven, The Netherlands) environmental scanning electron microscope, equipped with a 500 µm aperture gaseous secondary electron detector and a Peltier cooling stage. For the fermented emulsions with high dry matter content, a piece was cut with a scalpel (4 mm long × 3 mm large × 1.5 mm thick), and Cu-tape was used to fix the sample onto the crucible of the Peltier stage. For liquid emulsions and fermented emulsions with low dry matter content, 15 µL were deposited onto the crucible.

Condensation and evaporation conditions in the ESEM were obtained by imaging at a low sample temperature, i.e., 1.5 to 5 °C, while varying the water vapor pressure to reach relevant relative humidity (r.h.). In the first step, the sample environment was kept in a 100% r.h. environment. Then, the pressure was decreased to achieve a 50% r.h. and later a 15% r.h. The last step consisted of returning to a 100% r.h. condition.

In detail, the relative humidity changes were achieved as follows. In the first step, for the genuine dairy product samples, the sample environment was kept in a 100% relative humidity (r.h.) environment, i.e., with a water pressure of ~680–750 Pa and a temperature of ~1.5–2.0 °C. After the first series of imaging, the r.h. was decreased down to 50% (water pressure of 435 Pa and temperature of 5 °C) before a second series of images acquisition. A third series of imaging was performed at 15% r.h. (water pressure of 136 Pa and temperature of 5 °C) and, finally, the sample was brought back to the full hydration environment (water pressure of 870 Pa and temperature of 5 °C). For plant-based alternatives, the temperature was kept constant at 2.0 °C and the water pressure varied according to the requested relative humidity: 800 Pa for initial 100% r.h., 352 Pa for 50% r.h. and 105 Pa for the 15% r.h., then 703 Pa for the final 100% r.h.

All ESEM experiments were carried out under the following operating conditions: beam accelerating voltage of 10 kV, spot size of 3.0, sample working distances of 5.5–7.8 mm depending on the type of investigated samples, and 5–10 µs dwell time. Secondary electron images were acquired via the integrated imaging software (xT microscope control, xT plateform version 21.0.0.854) at magnifications from 500 X to 10,000 X. The images were taken from a single product, in at least two spots, whereby chosen images are displayed.

### 2.2. SAXS

SAXS experiments were performed using a laboratory-based SAXS/WAXS/USAXS beamline KWS-X (XENOCS XUESS 3.0 XL Garching Version) at JCNS MLZ. The instrument is equipped with a D2+ Excillum MetalJet X-ray source operating at 70 kV and 3.57 mA with Ga-Kα radiation (wavelength λ = 1.314 Å). Samples were measured in a glass capillary (1.5 mm ID) that was kept at 5 °C or 25 °C in a temperature-controlled stage. The sample-to-detector distances were from 0.1 m to 1.70 m, which cover the scattering vector q range from 0.003 to 5 Å^−1^ (Q is the scattering vector, Q = (4π/λ)sin(θ), 2θ is the scattering angle). The scattering patterns were obtained with a short exposure time of 60 to 600 s. Selected samples were measured by Bonse–Hart USAXS to extend the Qmin up to 0.0002 Å^−1^. The SAXS patterns were normalized to an absolute scale and azimuthally averaged to obtain the intensity profiles, and the solvent background was subtracted. Each SAXS curve was obtained from a single product.

The SAXS data was fitted with the Beaucage model [28]. This model identifies the peaks and shoulders in the SAXS curves, which simplifies the SAXS curve interpretation. These peaks and shoulders may be linked to structural components within the food matrix.

In detail, this Beaucage model fits well-separated length scales (Guinier scattering) and has a fractal scattering attached to it (with a fractal dimension *P*_i_). It allows relevant length scales and the fractality of the underlying objects to be obtained by combining a fit of the Guinier regime providing the radius of gyration, R_g,i_, of the relevant length scale, with a fit of the following power law decay, leading to the fractal dimension P_i_.

For hierarchically structured systems with separated length scales, this generic determination of the corresponding R_g,i_ and P_i_ can be added several times.

Peaks in the scattering curve stem from highly ordered regions with a crystal structure, such as crystallized fat molecules in the Q-region of 0.1–1 Å^−1^, or possible salt crystals with larger Q and atomic distances. The determination of the peak position is also included in our fitting procedure below. We assumed three distinct length scales (with radius *R*_g,i_) and a single correlation peak at *Q** for periodic structures. The length scale *d* for the periodic structures is related to *d* = 2π/*Q**. The overall model function for the SAXS intensity *I*(*Q*) reads as follows:$$I\left(Q\right)=\sum_{i=1}^{3}A_{i}\left(\exp\left(-\frac{1}{3}Q^{2}R_{g,i}^{2}\right)+\frac{P_{i}}{\Gamma (P_{i}/2)}\cdot {\left[\frac{{erf}^{3}(1.06QR_{g,i}/\sqrt{6})}{QR_{g,i}}\right]}^{P_{i}}\right)\;+A_{4}\exp\left(-\xi^{2}{\left(Q-Q^{*}\right)}^{2}\right)+b_{ckgr}$$

We also introduce the amplitudes *A_i_* and the correlation lengths ξ of the periodic structures. Γ(x) is the gamma function, erf(x) is the error function, and *b*_ckrg_ is the background. The first term in the sum, where the exponential function determines the radius of gyration of each hierarchical level of length scales. The second term is the power law decay and hence the fractality of this level. The contribution with amplitude A4 describes the correlation peak from crystalline structures. The small-angle scattering intensity I(Q) is usually converted to the instrument-independent macroscopic scattering cross-section dΣ(Q)/dΩ in units of cm^−1^, as shown later in the figures of the scattering curves. The macroscopic scattering cross-section describes the probability of the incoming beam (here the X-ray beam) being scattered by the sample into the direction defined by the scattering vector Q.

## 3. Results and Discussion

### 3.1. ESEM

Using ESEM, real-space information for dairy products and plant-based alternatives is shown for the categories of liquid emulsion and fermented emulsion with low and high dry matter content (Figure 1 and Figure 2). Using a secondary electron (SE) detector allows us to obtain information on the topography of the samples at the air (vacuum) interface in their wet state, after evaporation of a small amount of water, as described in Section 2.1. For most of the samples, we observe characteristic globular structures in the micrometer size range. The extent to which these structures deform and spread at the air (vacuum) interface reveals information about their internal cohesive and mutual adhesive strengths.

All dairy samples in Figure 1 reflect the structure of a two-phased system. The fine and homogenously dispersed oil phase and the continuous water phase with the respective solubles are especially visible in Figure 1a,c,d. For UHT milk shown in Figure 1a,d, we observe polydisperse, irregular-shaped aggregated structures in the size range of 0.5–5 µm, corresponding to milk fat globules that have aggregated at the air/liquid interface.

The structure of the liquid two-phased system, i.e., milk (Figure 1a,d), is strongly altered via coagulation during the fermentation step and via separation during further processing steps for the fermented emulsions with low and high matter content (Figure 1b,c,e,f). The yogurt sample shown in Figure 1b,e consists of well-defined spherical globules in the size range of 0.5–2 µm, which keep their shape as well as their aggregated porous matrix structure at the air/gel interface. This indicates sufficient cohesion of the globules to keep their shape, as well as sufficient adhesion between the globules to stabilize the three-dimensional porous matrix structure. Since we observe that UHT milk and yogurt have the same scaling of their X-ray scattering curves at low q (see Figure 3a), we conclude that the more irregular structures observed for UHT milk, compared to yogurt, are a result of wetting and adhesion to the air/liquid interface, and that in the bulk of the sample, the UHT milk globules are similar in shape and well stabilized, as is the case for the yogurt sample. Further, the globular structures in the yogurt system may also be influenced by the protein network that encapsulates the milk fat globules.

The resulting lipid and protein networks for yogurt are less dense than for Gouda, which is clearly linked to the lower dry matter, fat, and protein content of yogurt (see Table 1). The Gouda sample sustains a connected globule matrix structure with a solid-like appearance and a quite dense matrix, as shown in Figure 1f at high magnification.

However, ESEM is not commonly used to investigate the structure of dairy products in comparison to SEM, CLSM, or TEM [11]. The microstructure shown in the CLSM images of milk and cheese [11] is comparable to the respective microstructures in Figure 1.

Such a two-phased system is not identified for the plant-based alternatives in Figure 2. The plant-based alternatives may be categorized into more complex Pickering emulsions, as earlier indicated by Sarkar and Dickinson [29]. Further, the high contents in protein or long-chain carbohydrates for all plant-based alternatives (Table 1) seem to glue the single components together, causing a coarser and more inhomogenous structure than that in Figure 1. Such an increase in coarser and inhomogenous structure was also identified for plant-based cheese products based on coconut fat, sunflower oil, pea protein, and starch, analyzed via CLSM [30].

In detail, the plant-based liquid emulsion (Figure 2a,d) shows a coarser structure with sporadic suspended materials in comparison to the dairy liquid emulsion. We observe small globules in the 0.2–0.5 µm size range, in agreement with the results of the SAXS analysis (see Table 2), whereby globules are agglutinated with a polymer-like film and fibrils most likely from starch and fibers.

The “fermented” plant-based emulsion with low dry matter content (Figure 2b,e) resembles the structure of dairy yogurt in terms of coagulated spheres in the size range of 0.5–3 µm sticking together to form a dense, stable matrix. In contrast, the “fermented” plant-based emulsion with high dry matter content (Figure 2c,f) shows a rather inhomogenous structure, especially in comparison to Gouda (Figure 1c,f). We observe globules in the size range of 0.5–5 µm that appear to be surrounded by voids that have been formed by the evaporation of water and embedded in a continuous solid matrix. A recent publication [31] introduces some schematic drawings of plant-based cheese products, such as fermented3-CC and fermented2-C in Figure 2. These drawings strengthen our observation of very complex structural systems with embedded oil droplets, soluble carbohydrates, and a continuous protein network [31].

All these observations strongly indicate the structural differences between dairy products and plant-based alternatives. These differences are clarified and concretized within the length scale differentiating SAXS data in Section 3.2.

### 3.2. SAXS

SAXS showed the ensemble-averaged structure of the investigated samples. Structural differences within and between dairy products and plant-based alternatives are identified in Figure 3 and Table 2.

A high Q-value equals small structures, and vice versa, a low Q-value equals large structures. The size of each feature (peak or shoulder) may be approximated with 2π/Q. Peak-like feature A in Figure 3a,b resembles lamellar structures from, e.g., fat crystals. Shoulders and very broad shoulders, such as features B, C, and D in Figure 3a,b, are linked to less ordered structures, e.g., casein micelles. The decay of the curve also indicates some details about the features, which are explained later for each curve. The identification of the peaks and shoulders is simplified in Figure 3, with the legend listing all features. If a feature is strongly visible within the curve, the curve includes the abbreviation of the feature; otherwise, a slight indication of the feature is discussed within the text.

The scattering curves were also modeled with the Beaucage model [28]. This model helps to identify the features and their size as described in Section 2.2, leading to the three separated length scales with their R_g_ (radius of gyration) from the fit and the position of the correlation peak (size 4 in Table 2).

For all dairy products, the biggest structures of >1000 nm (size 1, Table 2, and incline in the curve at low Q in Figure 3a) are related to milk fat globules, which may have a diameter of 4000–5000 nm [11]. For milk, yogurt, and cream cheese, the structures with approx. 100 nm (size 2, Table 2, and feature D in Figure 3a) are related to casein micelles [6], and sizes of about 2 nm are linked to colloidal calcium phosphate (CCP) nanoclusters (size 2, Table 2, and feature B in Figure 3a) [7]. For all cheeses, the correlation peak with approx. 2π/Q= 4.2 nm (size 4, Table 2, and feature A in Figure 3a) is linked to lamellar structures of milk fat triacylglycerides with a double chain length [12]. The smallest length scales (size 3, Table 2) are related to nearly atomistic distances that arise from transient glassy states of the fat molecules [32,33]. Qualitatively, they exhibit typical features of an amorphous material: a main peak at Q = 1.4 Å^−1^, exhibiting a width AQ of roughly AQ = 0.4 Å^−1^, and a second peak at about Q = 2.9 Å^−1^ with AQ = 0.8 Å^−1^.

In detail, the features in each curve are described and connected with the ESEM images, starting from the curve on top and the low Q-range. For the scattering curve of the UHT milk, we observe, at the lowest *Q,* an *I(Q)*~*Q*^−2^-decay towards a shoulder at *Q** = 0.0025 Å^−1^, with a subsequent near Porod-*Q*^−4^-decay expected for the oil/water interface. This structural feature at *Q** corresponds to casein micelles with sizes of approx. 100 nm (Table 2, size 2). The observed second shoulder at *Q*** = 0.07 Å^−1^ corresponds to colloidal calcium phosphate (CPP) nanoparticles with a size of 2.3 nm diameter (Table 2, size 3). There are no apparent structural features at higher *Q*-values. Therefore, for UHT milk, we observe three structural hierarchies: the milk fat globules in the µm range as observed by ESEM, the casein micelles on length scales of approx. 100 nm, and the CCP nanoparticles on length scales of approx. 2 nm.

For yogurt, we also observe the *I(Q)*~*Q*^−2^-decay towards a shoulder at *Q** = 0.002 Å^−1^, with a subsequent near Porod-*Q*^−4^-decay, indicating the presence of slightly larger casein micelles. The shoulder at *Q*** is nearly absent, indicating the near complete dissolution of the CPP nanoparticles in the acidic environment. There is already a very weak peak at *Q**** = 0.14 Å^−1^ due to partial fat crystallization. Therefore, for yogurt, we observe four structural hierarchies: fat globules in the µm range as observed by ESEM; casein micelles on length scales of approx. 100 nm; a small fraction of CCP nanoparticles; and a small contribution from fat crystals with characteristic length scales or unit cells of 4.5 nm.

We observe that for cream cheese, the structural feature of the casein micelles has vanished into a mere change of slope from *I(Q)*~*Q*^−3^ to *Q*^−4^-scaling at *Q** = 0.002 Å^−1^, indicating disintegration of the casein micelles into a dense network of globules exhibiting a Porod *Q*^−4^-scaling over a large range of *q*-values. CPP nanoparticles are not observed anymore. At *Q**** = 0.14 Å^−1^ we observe a slightly large peak compared to the yogurt sample due to fat crystals. Thus, for cream cheese, we observe three hierarchies, with fat globules, aggregated casein micelles, and fat crystals.

For Camembert, the structural feature of the casein micelles has completely vanished, such that the low-q scattering is dominated by Porod *Q*^−4^-scattering of the fat globules. We observe a new structural feature characterized by a pronounced shoulder emerging at *Q*’ = 0.03 Å^−1^, presumably due to calcium–casein clusters with sizes of 8 nm (Table 1, size 2) [34], followed by a pronounced peak at *Q**** = 0.14 Å^−1^ due to fat crystals. Therefore, for Camembert, we observe three hierarchies, including fat globules, calcium–casein clusters, and fat crystals.

For Gouda, we observe a very similar scattering curve compared to Camembert. The only difference is that the shoulder at q′ has flattened and shifted slightly to lower q, indicating a slight increase in the size of the calcium-casein clusters, with partial disintegration to form a network structure. For the hard cheese, this trend is continued, with a further shift of Q′ to lower Q with *Q*′ = 0.01 Å^−1^. According to [34], this size increase of the calcium-casein clusters causes the increase in elastic modulus from Camembert and Gouda to the hard cheese. For all three cheeses, there is a pronounced peak at *Q**** = 0.14 Å^−1^. Thus, for these three cheeses, we observe three structural hierarchies: fat globules, aggregated calcium-casein clusters, and fat crystals.

For the plant-based alternatives (see Figure 2 and Figure 3), the structures are coarser and more inhomogenous [31], which makes the identification of the components more difficult. We do not observe any structural features on the length scales of the casein micelles. Instead, we observe a rather featureless *I(Q)*~*Q*^−2^ to *Q*^−4^-scaling over a large range at low *Q*. From the scattering experiments, we would attribute the largest structures (size 1, Table 2, and feature C in Figure 3b) to starch, which may be gelantized (100–300 nm), and potentially large protein particles [29,35]. For the liquid emulsion systems (drink 1 and drink 2), the starch and protein particles may function as pickering stabilizers at the oil/water interfaces, which may adsorb at the interface, form a network in the water phase, or function as a bridge between the interfaces [29].

For both drinks, we observe a weak, but characteristic *q*^−1^-scaling at high *Q* characteristic for persistent filamentous polymer structures most likely derived from starch and fibers. For drink 2 we observe a pronounced shoulder at *Q*″ = 0.06 Å^−1^ characteristic for structures with radii of ca. 5 nm (Table 2, size 2, and feature B in Figure 3b), which may be linked to oat protein. The most dominant protein fraction is 12S globulin (hexameric structure with up to 320 kDa) with 50–60% of the oat protein content [36]. In general, this feature B should also be visible in the curve of drink 1, since both drinks contain oat proteins. However, drink 1 contains pea and oat protein, which may cause a loss in the clear detection of this feature for drink 1.

These network structures tend to get bigger during gel formation via yogurt, cream cheese, and cheese production [31]. For the fermented yogurt, we hardly observe any clear structural features. The only characteristic is a change from a near Porod *Q*^−4^-scaling until *Q* = 0.06 Å^−1^, where it changes into a very weak decay. This feature is in the same size range as feature B for drink 2. Thus, it may also be linked to oat proteins, which are also a dominant protein fraction in this product beside soy protein (feature B in Figure 3b). In addition, the main fraction of soy protein, 11 S globulin, is comparable in size to the 12 S globulin in oat [37].

In a previous publication, the investigated cheese and cream cheese alternatives (fermented2-CC, fermented3-CC, fermented1-C, and fermented2-C) were classified as fractionation route cheeses [31]. Thus, these products are composed of preprocessed ingredients like protein isolates and isolated starch. The preprocessing affects the structure of these products strongly, since the raw materials have already experienced phase transitions, such as starch gelantization or protein denaturation and aggregation before the processing of the cheese itself [31]. For the three fermented cheese-like structures, there is a structural feature around *Q* = 0.06 Å^−1^, corresponding to structures in the size range of ~ 6 up to 13 nm (size 2, Table 2). We attribute this to protein structures for the curves of fermented3-CC and fermented2-C (feature B in Figure 3b). Again, peaks arise from well-aligned fat molecules (size 4, Table 2, and feature A in Figure 3b) from rapeseed oil [38] and 3.4–3.6 nm from coconut oil [39]. The smallest length scales (size 3, Table 2) are related to transient glassy states [32,33].

## 4. Conclusions

An overview of the multiscale structure of commercially available dairy and plant-based emulsions was presented. This publication serves as a first glimpse into the multiscale structure of complex food systems composed of protein, carbohydrates, and fat, among other components. Combining ESEM in real space and SAXS for an ensemble averaged view over a large range of length scales in reciprocal space provided the relevant structural features in these food emulsion systems. For dairy products, the identified structural features were milk fat globules, casein micelles, CCP nanoclusters, and milk fat crystals. With increasing dry matter content in dairy products, milk fat crystals were identified, and casein micelles were non-detectable due to the protein gel network in cheese. For the more inhomogeneous plant-based alternatives, fat crystals, starch structures, and potentially protein structures were identified. Details about the emulsion and gel structure of plant-based stabilized systems are still unknown and need to be further elaborated since the structure affects sensory aspects, such as mouthfeel and the texture of food products.

To clearly understand the mechanisms, an approach with model systems will serve as a first instance. This approach will be broadened with increasing protein variety and system complexity. A detailed understanding is important for further developments in the field of emulsions and their derived products, especially in view of exploiting plant proteins, which have, so far, been less focused on for dairy products. Such a structure-based comparison will lead to improved understanding and enhanced product development of plant-based alternatives in the future, since they symbolize the climate-neutral future of human nutrition.

## Figures and Tables

**Figure 1 foods-12-02021-f001:**
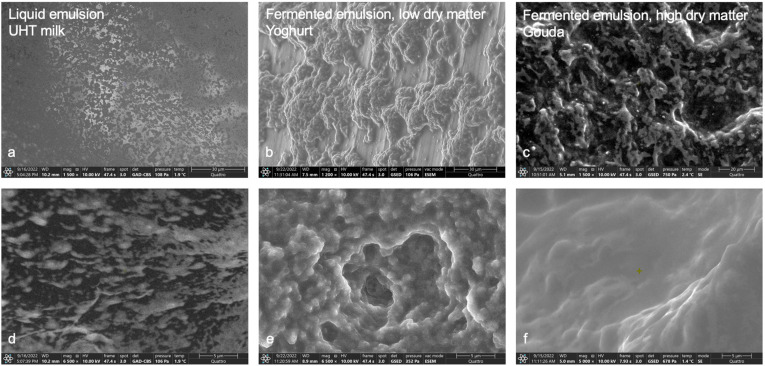
ESEM images of dairy products for each category: UHT milk as a liquid emulsion (**a**,**d**), yogurt as a fermented emulsion with low dry matter content (**b**,**e**), and gouda as a fermented emulsion with high dry matter content (**c**,**f**), captured with low (first row) and high magnification (second row).

**Figure 2 foods-12-02021-f002:**
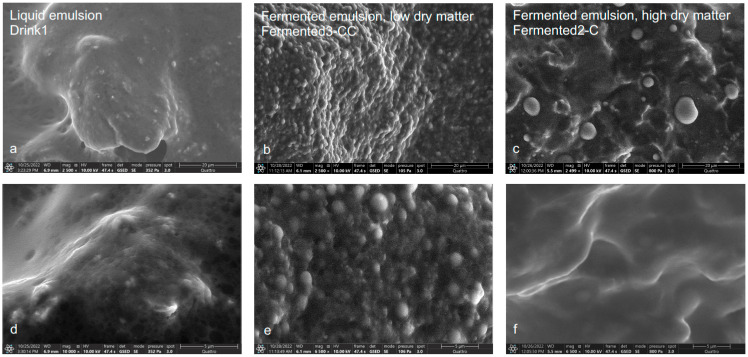
ESEM images of plant-based alternatives for each category: drink1 as a liquid emulsion (**a**,**d**), fermented3-CC as a fermented emulsion with low dry matter content (**b**,**e**), and fermented2_C as a fermented emulsion with high dry matter content (**c**,**f**), captured with low (first row) and high magnification (second row).

**Figure 3 foods-12-02021-f003:**
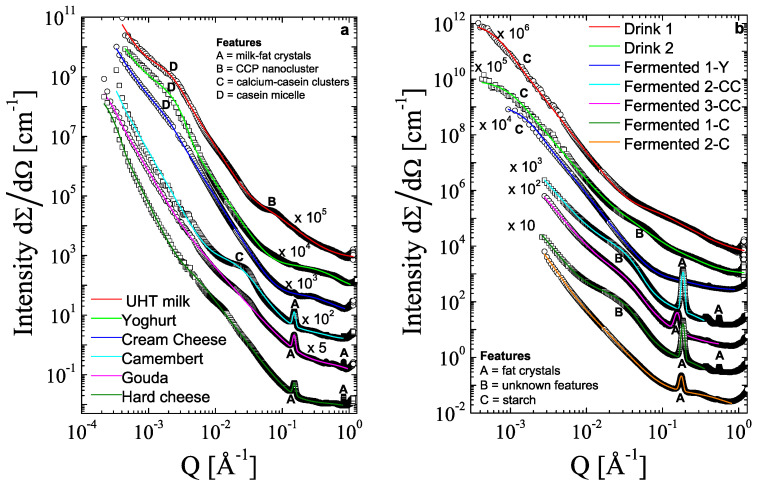
SAXS data of dairy products (**a**) and plant-based alternatives (**b**). For dairy products: UHT milk is the liquid emulsion; yogurt and cream cheese are the fermented emulsion with low dry matter content; and camembert, gouda, and a hard cheese are the fermented emulsion with high dry matter content. For the plant-based alternatives: two liquid emulsions called drink 1 and 2; three fermented emulsions with low dry matter content categorized as yogurt (fermented 1-Y) or crème cheese (fermented 2-CC, 3-CC); and two fermented emulsions with high dry matter content categorized as cheese (fermented 1-C, 2-C).

**Table 1 foods-12-02021-t001:** Information summarized from the product declaration. Overview of dairy products and plant-based alternatives with respective fat, saturated fat, carbohydrate, sugar, protein, and salt content (in %) according to the declaration of commercial products.

Category	Product	Fat (%), of Which Is Saturated	Carbohydrates (%), of Which Is Sugar	Protein (%)	Salt (%)
Liquid emulsion	**UHT milk**	3.5, 2.4	4.8, 4.8	3.3	0.1
Fermented emulsion, low dry matter	**Yogurt**	3.5, 2.4	5.1, 5.1	4.4	0.2
	**Cream cheese**	17.0, 11.0	3.0, 3.0	8.5	0.3
Fermented emulsion, high dry matter	**Camembert**	32.0, 22.0	0.5, 0.5	17.0	1.4
	**Gouda**	28.0, 18.8	0.5, 0.5	23.5	1.5
	**Hard cheese**	29.7, 19.6	0.0, 0.0	32.4	1.6
Liquid emulsion	**Drink 1**	3.5, 0.4	5.7, 0.0	0.7	0.1
	**Drink 2**	3.0, 0.3	7.1, 3.4	1.1	0.1
Fermented emulsion, low dry matter	**Fermented 1-Y**	1.9, 0.4	3.4, 1.0	3.8	0.3
	**Fermented 2-CC**	23.0, 21.0	8.0, 0.0	0.0	1.2
	**Fermented 3-CC**	20.0, 8.0	9.8, 3.6	3.2	0.7
Fermented emulsion, high dry matter	**Fermented 1-C**	29.0, 26.0	11.0, 0.0	0.0	1.7
	**Fermented 2-C**	19.0, 17.0	24.0, 0.5	0.5	1.9

**Table 2 foods-12-02021-t002:** Structural features of SAXS data of dairy products and plant-based alternatives according to Beaucage modeling.

Product	Size 1 [nm]	Exponent	Size 2 [nm]	Exponent	Size 3 [nm]	Exponent	Size 4 (Periodic) [nm]
UHT milk	>1000	4 fix	96 ± 1.7	3.59 ± 0.01	2.34 ± 0.07	1.99 ± 0.02	--
Yogurt	>1000	3.33 ± 0.01	95.1 ± 1.6	5.23 ± 0.01	0.37 ± 0.04	1.9 ± 1.0	--
Cream cheese	>1000	3.50 ± 0.02	123 ± 4	3.71 ± 0.05	0.39 ± 0.01	2.4 ± 0.3	--
Camembert	>1000	4 fix	7.7 ± 0.1	3.17 ± 0.03	0.39 ± 0.02	large	4.18 ± 0.01
Gouda	>1000	3.82 ± 0.01	40 ± 3	2.39 ± 0.01	0.15 ± 0.03	large	4.19 ± 0.01
Hard cheese	>1000	4.59 ± 0.01	40.4 ± 0.9	2.93 ± 0.01	0.17 ± 0.17	small	4.17 ± 0.01
Drink 1	321 ± 8	3.51 ± 0.01	1.68 ± 0.05	1.72 ± 0.03	--	--	--
Drink 2	197 ± 4	3.22 ± 0.02	4.8 ± 0.5	2.3 ± 0.2	0.61 ± 0.06	2.5 ± 0.2	--
Fermented1-Y	130 ± 1	3.44 ± 0.01	--	--	0.71 ± 0.01	2.2 ± 0.1	4.03 ± 0.01
Fermented2-CC	>100	3.10 ± 0.06	6.4 ± 0.1	4.2 ± 0.1	0.66 ± 0.06	2 fix	3.38 ± 0.01
Fermented3-CC	>100	3.22 ± 0.02	6.39 ± 0.06	3.62 ± 0.02	0.71 ± 0.04	2 fix	4.03 ± 0.01
Fermented1-C	>100	3.40 ± 0.06	6.6 ± 0.1	3.74 ± 0.04	0.72 ± 0.06	2 fix	3.39 ± 0.01
Fermented2-C	>100	3.89 ± 0.05	13.3 ± 0.7	2.79 ± 0.02	0.311 ± 0.003	2 fix	3.55 ± 0.01

## Data Availability

The data presented in this study are available on request from the corresponding author.

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
