# Peer review of "Multiscale Structural Insight into Dairy Products and Plant-Based Alternatives by Scattering and Imaging Techniques"

_foods, 2023, doi:10.3390/foods12102021_

Round 1

Reviewer 1 Report

Dear Editor and Authors,

I send you my review about the article “Multiscale structural insight into dairy products and plant-based alternatives by scattering and imaging techniques”.

The scope of the Article was to investigated the multiscale structural insights of dairy products and plant-based alternatives via small angle X-ray scattering in association with environmental scanning electron microscopy.

In my opinion, although the Article is well written and well structured, it show also some lacks that I reported below.

In general, the product “IT Y8020 CE” is named Parmigiano Reggiano PDO cheese and not “Parmesan”. Parmigiano Reggiano is a trade mark and, for this reason, it name should be reported as it is was recorded and it not should be translate. This to avoid confusing readers.

The introduction was well written and it well reported the literature of this topic but, the aim of the research, should be better explained. 

Moreover, in the introduction, the sentence from line from line 56 to line 58 should be shift in the conclusions.

The paragraph materials and methods was well written and well structured, but it should be better explained the difference among the number of samples collected for each type of product and the number of replicate of analysis performed on each samples.

Furthermore, in the chapter Materials and methods should be reported the methods of analysis of the chemical parameters determined on the products and reported in the table 1.

Moreover, in the chapter Materials and methods should be added a paragraph of data analysis were the Authors should report how them analysed data collected.

In addition in this paragraph should be reported the sentences form line 214 to line 224.

The results were well shown and their discussion was complete, but the table 2 is divided in two different pages. In my opinion to facilitate the understanding of the data by the readers the tables should be placed in a single page.

Finally, the conclusions were adequate to the data showed and to the aim of the research, but, in my opinion, it should better stress the relevance of the findings of this research.

Best regards

Author Response

Thank you very much for your valuable comments. 

We included your suggested changes. 

  1. The Parmesan is called hard cheese, and is introduced correctly.
  2. We included the indicated changes in the introduction, and material and methods section, as well as in the results and conclusion section. 

I hope that you agree with the changes. Since the other reviewers pointed at some other issues as well, the changes in the manuscript are stronger than indicated in your revision. 

Please take a look into the attachment, thus revised manuscript. 

Reviewer 2 Report

The study is very specialized and the work should be simplified for the average reader. The main objective of publishing is to share ideas and explain how they could be useful for further applications. This is not precisely clear in this study.

There is no assurance of reproducibility of data or statistical analysis. Also, the discussion is all about observational descriptive information that cannot be observed by a regular reader. 

I added some comments in the PDF file for your consideration.

it will require moderate language review. I tried to help up to a certain page and then left it to the authors to sort it out.

Author Response

Thank you very much for your valuable comments. I am a food scientist, and aim to spread the usefulness of SAXS and SANS to the food community. Thank you for your review! 

We clarified the aim of the work in the introduction section and conclusion section, which links to the usefulness of the results. 

We included some paragraphs to bridge the classical food science and scattering point of view. We also changed the figures to ease the interpretation. We introduced the scattering details in a very easy way. We strongly hope that this improves the understanding.

We included the statistical assurance for the SAXS data in Table 2. We also improved the section 2.2, to improve the understanding of the SAXS curve fitting model - Beaucage model. 

We improved the language issues as well.

We strongly hope, that you agree with our revised manuscript. 

Reviewer 3 Report

The manuscript describes the original research on the multiscale structure of commercially available milk- and plant-based emulsions. The manuscript contains a well-prepared section " 1. Introduction" describing the purpose of the research. However, the authors did not describe the usefulness of the obtained results, nor did they attempt a statistical analysis of the obtained results, comparisons between the examined samples, or looked for a correlation of the results with other parameters of the examined market products, albeit only the basic composition or physico-chemical or texture analysis. In my opinion, these aspects should be completed in the manuscript. Otherwise, the results obtained will have no scientific or applied value.

Author Response

Thank you very much for your valuable comments. 

We revised the manuscript and improved the presentation of the aim in the introduction section. The aim, thus the usefulness of the results, is also addressed in the abstract and conclusion. 

In the first version, we compared the samples, and correlated the ESEM and SAXS results, already. I restructured the text a bit, and hope that it gets more clear. The first paragraph about dairy products and the first paragraph about plant based emulsions in each section (ESEM and SAXS) starts with a comparative and generic paragraph. The description is getting more and more specific for each product, but we also compare between the samples throughout the discussion and results part (e.g. line 212, line 218, line 226, line 252-254, line 281-291, line 339 to 341).  We also bridge the ESEM and SAXS results which I classify as correlation of results e.g. line 300-303, line 309 to 311, since we identify the fat globules in ESEM and SAXS. We included a longer discussion and comparison of results in line 357-372. The comparison of the results is included in the abstract and conclusion, now. 

The statistics of the SAXS data is included in Table 2. 

This study functions as a first base, and needs to be extended to model and more complex systems to understand the mechanisms causing the food structure. When we achieved that, I would go into a comparison with a sensory study with a proper panel, and texture analysis.

However, I did not find any study comparing such a big data set with SAXS and ESEM with such a broad dry matter content range as indicated in the introduction. For dairy products, the SAXS data interpretation is easy, but for plant based products the lack in studies complicates the data interpretation and complicates to broaden the understanding. We addressed this issue in the conclusion.

I hope that you agree with the changes in the revised manuscript. 

Round 2

Reviewer 3 Report

The current version of the manuscript is much better than the previous one. I find no other comments or feedback on this test and accept the current version for publication in FOODS journal.